# Molecular Analysis of the Global Population of Potato Virus S Redefines Its Phylogeny, and Has Crop Biosecurity Implications

**DOI:** 10.3390/v15051104

**Published:** 2023-04-30

**Authors:** Şerife Topkaya, Ali Çelik, Adyatma Irawan Santosa, Roger A. C. Jones

**Affiliations:** 1Department of Plant Protection, Faculty of Agriculture, Tokat Gaziosmanpasa University, Tokat 60250, Turkey; 2Department of Plant Protection, Faculty of Agriculture, Bolu Abant İzzet Baysal University, Bolu 14030, Turkey; 3Department of Plant Protection, Faculty of Agriculture, Universitas Gadjah Mada, Yogyakarta 55281, Indonesia; 4The UWA Institute of Agriculture, The University of Western Australia, Crawley, WA 6009, Australia

**Keywords:** potato, PVS, Turkish provinces, phylogeny, molecular characterization, recombination, genetic diversity, population genetics, gene flow, gene transcript analysis, incursions, biosecurity

## Abstract

In 2020, 264 samples were collected from potato fields in the Turkish provinces of Bolu, Afyon, Kayseri and Niğde. RT-PCR tests, with primers which amplified its coat protein (CP), detected potato virus S (PVS) in 35 samples. Complete CP sequences were obtained from 14 samples. Phylogenetic analysis using non-recombinant sequences of (i) the 14 CP’s, another 8 from Tokat province and 73 others from GenBank; and (ii) 130 complete ORF, RdRp and TGB sequences from GenBank, found that they fitted within phylogroups, PVS^I^, PVS^II^ or PVS^III^. All Turkish CP sequences were in PVS^I^, clustering within five subclades. Subclades 1 and 4 were in three to four provinces, whereas 2, 3 and 5 were in one province each. All four genome regions were under strong negative selection constraints (ω = 0.0603–0.1825). Considerable genetic variation existed amongst PVS^I^ and PVS^II^ isolates. Three neutrality test methods showed PVS^III^ remained balanced whilst PVS^I^ and PVS^II^ underwent population expansion. The high fixation index values assigned to all PVS^I^, PVS^II^ and PVS^III^ comparisons supported subdivision into three phylogroups. As it spreads more readily by aphid and contact transmission, and may elicit more severe symptoms in potato, PVS^II^ spread constitutes a biosecurity threat for countries still free from it.

## 1. Introduction

Potato virus S (PVS, genus *Carlavirus*) is one of the most prevalent viruses infecting potato (*Solanum tubersosum*) crops globally. It was reported first in 1952 in Europe [1] and subsequently in all major potato production areas across the world [2,3,4,5,6,7]. Following potato virus Y (PVY, genus *Potyvirus*), potato leafroll virus (PLRV, genus *Polerovirus*) and potato virus X (PVX; genus *Potexvirus*), PVS is considered the fourth most important viral pathogen affecting potato production worldwide [3,4,7]. It generally causes mild symptoms or asymptomatic infection in potato foliage and only minor yield losses. However, when severe PVS strains are present, tuber yield losses can reach 20% and tuber size is diminished. Moreover, PVS commonly occurs in mixed infections with other common potato viruses, such as PVY, PLRV, PVX and potato virus A (PVA, genus *Potyvirus*). This co-infection can increase PVS virion content within infected plants thereby intensifying symptom expression and causing greater tuber yield losses than that occurring with individual infections [3,4,7,8,9,10,11]. As PVS is transmitted easily by foliage contact, and often causes subtle symptoms or asymptomatic infection when present on its own, managing it in healthy seed potato multiplication schemes is problematic. This is especially so in potato seed schemes that rely on visual crop inspections for symptoms without complementing such inspections with regular virus tests on randomly collected leaf samples. This means that PVS is often the first virus to invade healthy seed potato stock schemes. Thorough sample testing in the beginning phases of clonal selection or healthy min-ituber multiplication combined with phytosanitary control measures that minimize contact transmission are required to detect and remove this virus contamination [3,4,7,8,10].

As with other members of genus *Carlavirus*, PVS has filamentous particles containing a single-stranded, positive-sense RNA genome approximately 8.5 kb in length. This genome is composed of 34 kDa coat protein (CP) and contains a 5′ cap structure, an open reading frame (ORF) encoding an RNA-dependent RNA-polymerase (RdRp), the triple gene-block proteins (TGBp1-3) involved in virus cell-to-cell movement, and a NABP (cysteine-rich nucleic-acid-binding protein) [12]. Two PVS strains were recognized in early biological studies, PVS^O^ (ordinary) and PVS^A^ (Andean). These strains were distinguished depending on whether they remained in inoculated leaves of *Chenopdium quinoa* (PVS^O^; O = Ordinary) or invaded this host systemically (PVS^A^; A = Andean) [13,14,15,16].

Complications arose later, however, when PVS isolates were sequenced, and this same nomenclature (PVS^O^ and PVS^A^) was retained to distinguish the two major phylogroups found following phylogenetic analysis. This problem arose because biologically defined PVS^A^ isolates also occurred within major phylogroup PVS^O^ [17] and biologically defined PVS^O^ strain isolates occurred within major phylogroup PVS^A^ [6,18]. The term PVS^CS^ (CS = *Chenopodium* systemic) was suggested to distinguish PVS^A^ isolates from other isolates within major phylogroup PVS^O^ [17]. Subsequently, Cox and Jones [18] suggested the terms PVS^O-CS^ (CS = *Chenopodium* systemic) for major phylogroup PVS^O^ isolates which invaded this host systemically, and PVS^A-CL^ (CL = *Chenopodium* localized) for major phylogroup PVS^A^ isolates that remained localized within inoculated leaves. Moreover, when several Andean PVS isolates from *S. phureja* in Colombia were sequenced, an additional major phylogroup was found [19,20,21]. This was named PVS^RVC^ [5]. In the same year, Santillan et al. [6] reported findings on two further Colombian isolates from *S. phureja* and an Ecuadorian isolate from *S. tubersosum* ssp. *Andigena* that were within this same phylogroup, referring to it as a ‘second South American lineage’ rather than as PVS^RVC^. Ten additional PVS isolates from five countries in the potato’s Andean domestication center (Bolivia, Chile, Colombia, Ecuador, and Peru) all fitted within the predominantly Andean phylogroup, previously called PVS^A^, which they referred to as the ‘first South American lineage’. However, they retained the names PVS^O^ and PVS^A^ for the two strains defined biologically based on their respective inabilities or abilities to invade *C. quinoa* systemically. Santillan et al. [6] adopted their geographically based approach towards phylogroup nomenclature to avoid the confusion arising from using the same names (PVS^O^ and PVS^A^) for isolate groupings defined in two entirely different ways despite the lack of any relationship between them. To overcome the confusion caused by using names derived from biological properties or geography for plant virus phylogenetic groupings, Jones and Kehoe [22] had suggested replacing all the previous names of strains defined by biological or geographic differences within species phylogenetic group nomenclatures with Latinized numerals. Thus, according to this system, PVS^O^ and PVS^A^ would be retained for biologically defined PVS strains, but the names PVS^I^, PVS^II^ and PVS^III^ would replace the major phylogroup names PVS^O^, PVS^A^ and PVS^RVC^, respectively.

From their PVS phylogeny which included new PVS sequences from potato’s Andean region domestication center, Santillan et al. [6] concluded that their most important deduction was “amongst all three lineages (= PVS^I^, PVS^II^ and PVS^III^) from the root lead to existing populations infecting potato, only one (= PVS^I^) dominated the recent global adoption of potato as a major food crop”. After the Spanish invasion of Peru in 1532, PVS was introduced to Europe with potatoes during the “Colombian Exchange” period, which started in c.1570. However, the recent rapid diversification of PVS^I^ only commenced after the major introduction of new potato germplasm from the Andean region which occurred following the European potato famine of the 1840s caused by potato blight (*Phythopthora infestans*). Moreover, Duan et al. [5] concluded from their PVS phylogeny that Europe played a major role in dispersing this virus to other continents via multiple migration pathways. They also concluded that future studies on bigger data sets with broader geographic representation were necessary to obtain a more complete picture of PVS’s evolutionary history.

RNA viruses, such as PVS, have a strong potential to grow and adapt quickly to natural selective forces because of their enormous population size, proneness to quasispecies development, lack of genome proofreading systems, and rapid generation rates allowing the development of considerable genetic variation [23]. When genetic variation results in functional gain, the high frequency of mutation, recombination, and reassortment that occurs in viral genomes encourages generation of novel forms that quickly spread across the viral population [24]. Therefore, to control viral pathogens effectively, it is important to understand virus population structure and assess its diversity [25,26].

Potato is one of Turkey’s most important food crops, being grown in 14,799 ha of land and with a total annual production of 5,200,000 tons [27]. PVS presence in the country was established when potato samples were tested by ELISA using PVS antibodies [28,29,30,31] or reverse transcription-polymerase chain reaction (RT-PCR) with PVS specific primers [32,33,34,35]. Güner and Yorganci [29] found PVS in the potato production areas of Nevşehir and Niğde provinces, and Engür and Topkaya [34] reported it in the Tokat province. However, despite these studies suggesting widespread PVS occurrence, there is insufficient molecular information about Turkish PVS isolates. In this study, after initial surveys in which we collected PVS-infected leaf samples from Turkish potato fields and obtained the CP sequences of new isolates, we conducted a molecular analysis that redefines the phylogeny of the global PVS population. Our aims were (i) to provide a comprehensive overview of the phylogenetics of PVS using complete ORF, RdRp, and CP genes of different isolates to characterize its genetic diversity and evolutionary history, and (ii) to analyze Turkish isolate coat protein (CP) gene sequences to provide information about PVS biosecurity threats in Anatolia and neighboring Middle Eastern countries.

## 2. Materials and Methods

### 2.1. Surveys

During the 2020 summer growing season, 264 leaf samples were collected from potato fields in the Turkish provinces of Afyon, Bolu, Kayseri, and Nevşehir. They were stored at 4 °C for 1–2 days before use. Most of these samples were from plants showing virus-like symptoms, but a few were from symptomless plants.

### 2.2. RT-PCR and Sequencing

Total RNA was extracted from leaf tissue from each sample as described by Astruc et al. [36]. RT-PCR was performed on each sample individually. To create cDNA in the reverse transcription, a mixture containing 2.5 µL total RNA, 4 µL 5× RT-buffer, 0.5 µL dNTPs (25 mM), 0.25 µL hexamer primer (10 μM) and dH_2_O was incubated at 42 °C for 1 h. For PCR, a 25 μL mixture containing 2.5 μL of cDNA, 0.2 μL of 25 mM dNTPs, 2 μL of 25 mM MgCl_2_ and 5 μL of 5× green reaction buffer was used. Next, 0.5 μL of 10 μM of each of forward and reverse primers (5′-TGGGGAATCAGTCCGGCTAGTC-3′ and 5′-ACTGGACCTGCGCTTAGGCT-3′) were added to this mixture. To amplify the complete CP region of the PVS genome [37], 1.25-units GoTaq DNA polymerase (Promega, Madison, WI, USA), and sterile ultra-pure water were prepared. Amplification was performed as follows: initial denaturation for 5 min at 94 °C, 35 cycles of denaturation at 92 °C for 45 s, annealing at primer optimized temperatures 62 °C for 45 s, 72 °C for 1 min and final extension at 72 °C for 7 min. PCR products were then electrophoresed in 1.5% agarose gel stained with ethidium bromide. A single band of around 1100 bp was expected from positive samples. The PCR amplicons for 14 selected PVS positive samples were submitted to a commercial company (Macrogen, Singapore) to be sequenced bi-directionally using Sanger technology. Unfortunately, complete genomic sequencing was impossible, because of the limited research funding and facilities in the laboratory where the sequencing was performed.

### 2.3. Recombination and Phylogenetic Analysis

On 2 November 2022, complete genome sequences of 139 PVS isolates were retrieved from GenBank and aligned using the ClustalW algorithm included in the MEGA X software [38]. Then, their 5′ and 3′ ends were trimmed to create a ‘complete ORF’ alignment. Nine isolate sequences with recombination events detected by at least five of the RDP, MaxChi, GENECONV, BootScan, ChiMaera, 3Seq, and SiScan, and algorithms with Bonferroni-corrected *p* value of < 0.05 implemented in Recombination Detection Program (RDP) v.5.30 [39] were removed from the alignment. Then, the recombinant-free complete ORF alignment was subdivided sequentially to form alignments of RdRp, TGB and CP genes according to the reading frames of PVS RefSeq NC_007289. The complete CP sequences (885 nts) of the 14 novel isolates from this study, the 8 other Turkish isolates from Engür and Topkaya [34] and 73 other isolates with complete CP sequences from GenBank were added to the CP alignment of 130 sequences to create a 225-isolate dataset. Present in the ORF phylogenetic tree of Santillan et al. [6] but excluded from our analyses were the incomplete PVS^III^ sequences MF4I8029 (Ecuadorian *S. tuberosum* spp. *andigena* isolate with most of its CP gene missing), and JX683388 (Colombian *S. phureja* isolate with most of its RdRP gene missing). A list of isolates analyzed in our study is available in Appendix A.

Maximum-likelihood (ML) trees each based on complete ORF, RdRp, TGB or CP genes, were built using the best-fit Tamura–Nei parameter model [40] within MEGA X, with 1000 bootstrap replicates to determine the statistical significance of isolate clusters. The distance matrixes amongst the CP sequences of isolates tested at nucleotide (nt) and amino acid (aa) levels were analyzed using SDTv1.2 [41].

### 2.4. Population Structure

Estimation of population genetics-related parameters was performed for both complete ORFs and individual coding sequences (RdRp, TGB and CP) using DnaSP v.6.12.03 software [42]. The parameters calculated included number of haplotypes (*h*), haplotype diversity (*Hd*), number of variable sites (*S*), total number of mutations (*η*), average number of nt differences between sequences (*k*), nt diversity (per site) (*π*), and the acting selection pressure (*ω* = dN/dS). The calculations also included three neutrality tests with a window length of 100 sites and step size of 25 sites, Fu and Li’s *D** and *F** [43], and Tajima’s *D* [44]. To give insight into genetic differentiation and gene flow between the major PVS phylogroups, the parameters *K*S*, *KST**, *Z**, *Snn*, and *FST* (fixation index) metrics [45,46] were also calculated for complete ORFs and each of the coding sequences again using DnaSP v.6.12.03 software. The coefficient FST ranged between 0 (panmixia) and 1 (fully distinct populations) [45]. Therefore, a FST value > 0.33 indicates rare gene flow and expanding genetic separation amongst tested populations [47,48].

## 3. Results

### 3.1. PVS Occurrence and Sequences

The RT-PCR tests with PVS-specific primers detected this virus in 13% of the leaf samples collected (Table 1). The samples found infected mostly came from potato plants that exhibited foliage symptoms of mosaic, mottle, leaf curling and/or plant stunting. However, a few infected plants were asymptomatic, e.g., the infected potato plant in Afyon from which isolate SA12-5 was obtained (Figure 1A–E). The more severe virus symptoms most likely came from mixed infections caused by PVS and other common potato viruses. However, knowledge of other potato viruses that might have been present is lacking as the sequenced samples were not tested for the presence of other viruses. The PVS incidences amongst the samples from each province were as follows: Afyon (19%), Bolu (11%), Kayseri (9%), and Nevşehir (11%) (Table 1). Thus, the highest PVS incidence was in Afyon and the lowest in Kayseri. The best preserved PVS positive samples were selected for sequencing. Complete sequences of the CP region of 14 new PVS isolates were obtained (Afyon = 6, Bolu = 1, Kayseri = 3, Nevşehir = 4) and registered under accession nos. OP819684-OP819697 in the NCBI GenBank.

### 3.2. Recombination and Phylogenetic Analyses

Significant recombination events were detected using RDP5 analysis in different regions of the genomes of 9 out of 139 PVS isolates available in GenBank (Table 2). Five of these had been reported previously (AJ863510, LN851189, LN851192, LN851193 and LN851194) [6], but four others were new (KC430335, LC375227, MN689463 and MK096268). These recombinants came from Eastern Europe, East Asia and East Africa, and following were their countries of origin: Ukraine (3), Czech Republic (1), Poland (1), China (2), Japan (1) and Kenya (1). Their parents mainly belonged to phylogroup PVS^I^. However, MN689463 from Kenya had a minor PVS^II^ parent also from Kenya, and AJ863510 from the Czech Republic had a minor PVS^II^ parent from the Andean country of Bolivia. It had two recombination events, one at each end of its genome and both involving the same Bolivian minor parent. Additionally, LC375227 from Japan had Chinese recombinant sequence MK096268 as its major parent, and LN851194 from Poland had recombinant sequence LN851192 from Ukraine as its minor parent. With the exception of two Ukrainian recombinant sequences (LN851192 and LN851193) which had identical major and minor parents, and breakpoints at nucleotide sequence position 64–2751, all the minor parental nucleotide sequence positions were different. Thus, no recombination hotspots were revealed. None of the 10 recombinants were from Turkey or the potato’s Andean domestication center in South America.

The subsequent phylogenetic and population genetic analyses for complete ORF, RdRp, and TGB included only the 130 non-recombinant sequences.

Sixty-three GenBank isolates (three from Tokat and sixty others) were selected to represent PVS phylogroups I, II and III in nt and aa comparative analyses. According to SDT matrix analysis (Figure 2), the 14 new Turkish PVS isolates shared 94–99% nt and 97–100% aa identities with each other, 94–99% aa and 97–99% nt identities with the three Tokat isolates, and 79–99% nt and 93–100% aa identities with 60 other isolates from GenBank. The most genetically divergent new Turkish isolates were B13, B9, NP49, and SA125, which was confirmed by their phylogenetic placement (see next paragraph).

### 3.3. Phylogenetic Analysis

In our study, all of the complete ORF, RdRP, TGB and CP trees showed identical topographies in which PVS isolates were clustered into the same three major phylogroups found previously, PVS^I^, PVS^II^ and PVS^III^ (Figure 3 and Figure 4). However, during the intervening 4–5 years since the two most recently published PVS phylogenies [5,6], the addition of further new PVS sequences to GenBank has substantially altered the spectrum of geographical origins of the PVS sequences within PVS^I^ and PVS^II^. Our ORF, RdRp and TGB phylogenetic trees revealed that PVS^II^ now contains the following sequences: 15 Andean (Bolivia, Chile, Colombia, Ecuador, and Peru), one non-Andean South America (Brazil), 10 African (Kenya), two each from Australasia (New Zealand), East Asia (China) and Europe (the Netherlands), and one from Central Asia (Kazakhstan) (Figure 3A–C). Similarly, our PVS^II^ CP phylogenetic tree now contains the following sequences: 19 Andean (Bolivia, Chile, Colombia, Ecuador, Peru), two non-Andean South American (Brazil), 12 East Asian (China), 10 East African (Kenya), two each from Australasia (New Zealand) and Europe (the Netherlands), and one South Asian (India) (Figure 4). In our four different types of phylogenetic trees, PVS^II^ shares a basal node with PVS^III^ and all six sequences within PVS^III^ are Andean (Colombia) from *S. phureja.* All of the Turkish isolate CP sequences fitted within major phylogroup PVS^I^ within our CP tree (Figure 4). In addition, in all four of our phylogenetic trees (especially our CP tree), PVS^I^ now contains sequences from more continents (Figure 3 and Figure 4), primarily the ones from Kenya, Africa (45 sequences), and also ones from other major regions worldwide (South America, North America, Europe, Middle East, Central Asia, South Asia, East Asia, South East Asia, and Australasia). Notably, however, only four of these sequences are Andean (from Chile, Peru, and Ecuador), so PVS^I^ consists of an increasingly ‘rest of the world’ phylogroup.

The CP nt sequences from all the 22 Turkish isolates fitted within 5 different PVS^I^ subclades (Figure 4): (1) 8 Turkish isolates (PA3-3, PN14-3, B9, B13, NP49, PN5-2, PN3-6, and SA12-5) grouped closest to 5 isolates from the neighboring countries of Syria and Iran (AB364945, HQ875132, HQ875137, and MF773984-5) and 10 from Netherlands, Slovakia, Russia, Hungary, Germany and Slovenia (GU319951, MF418026, MF418022, MF346599, OL472247-9, LC511868, HF571059, LN794160, MH937416, and OM471986); (2) 4 Turkish isolates (PA3-2, PB5-4, PB6-2, and PB7-1) grouped closest to 1 isolate from the neighboring country of Iran (HQ875142) and 7 from Kenya (MN689446-7, MN689456-7, MN689481, MN689495, and MN689460); (3) B14 was closest to 1 from China (KU896945) and 1 from South Korea (U74375); (4) 8 isolates (B41, N52, NP80, Bo7, N73, Ka1-1, Ka1-7, and Ka1-14) grouped with 1 isolate from the neighboring country of Syria (AB364946) and 4 from the USA (FJ813513-14, JX183951, and JX183953); and (5) B20 was closest to 2 from Canada (MN950792–93). These results are consistent with 5 different PVS introductions to Turkey.

When the 22 Turkish isolates sequences were each mapped to the province they originally came from, Afyon contained the most diverse population with representatives of subclades 1, 3, 4 and 5 all present (Table 1 and Figure 5). Tokat and Nevşehir contained isolates from two subclades each, 1 and 4 for Nevşehir and 1 and 2 for Tokat. Kayseri and Bolu each only contained subclade 4. Thus, subclades 2, 3 and 5 each only occurred in one province whereas subclades 1 and 4 were the most widespread as they were found in three (subclade 1) or four (subclade 4) provinces.

### 3.4. Genetic Diversity and Selection Pressure Analyses

According to analyses based on their complete ORFs and RdRp, TGB and CP coding regions, major phylogroup PVS^II^ isolate sequences showed high divergence despite being relatively few in number (Table 3). The assigned *k* (nt differences between sequences) and π (nt diversity) values for the PVS^II^ population were the most obvious parameters as they were consistently higher than those of PVS^I^ and PVS^III^ across all regions studied. The highest *S* (variable sites) and *η* (mutation) values were exhibited by PVS^I^, closely followed by PVS^II^. The CP was the most conserved region probably because it is shorter than both of the RdRp and TGB regions. Different regions of the PVS genome were all under strong transcriptional selective pressure as shown by the low dN/dS ratio (ω < 0.2) obtained. Except in the RdRp analysis, PVS^II^ always received smaller ω values than PVS^I^. Overall, the CP had a lower ω value than other coding regions (Table 3).

### 3.5. Neutrality Tests

When three methods were applied in the neutrality tests of complete ORFs and RdRp, only for PVS^III^ were the values estimated for the TGB and CP coding regions consistently positive (Table 4). Additionally, in the analysis of complete ORFs and RdRp, the Fu and Li’s *D** test also gave positive values with PVS^II^ isolates. However, there was no statistically significant support for these differences, indicating there were insufficient data to draw strong conclusions for all populations.

### 3.6. Gene Flow and Genetic Separation amongst Populations

In comparisons between PVS isolate sequences belonging to all three major PVS phylogroups, DnaSP analysis estimated high and statistically significant *KS*, *KST*, and *Z* values (genetic differentiation statistics). According to all complete ORFs, RdRp, TGB and CP coding region comparisons, the *Snn* parameter (genetic differentiation) also always reached its maximum value (1.0000). Furthermore, all *FST* (coefficient of gene differentiation) values for the three phylogroup comparison were > 0.6 (Table 5). These results provided strong evidence that the division of PVS into three major phylogroups is justified.

## 4. Discussion

### 4.1. Research Highlights

Here, we provide new insights into the phylogeny of the global PVS population, and the genetic variation present within its local population in the Middle Eastern country of Turkey. We provide further evidence justifying the renaming of its three major phylogroups as PVS^I^, PVS^II^ and PVS^III^ rather than continuing with biologically or geographically based nomenclature, and confirm that the smallest phylogroup, PVS^III^, shares a basal node with PVS^II^. PVS^III^ remains entirely Andean in distribution, PVS^II^ has spread further outside the Andes, especially to East Africa and East Asia, and PVS^I^ now occurs worldwide. Indeed, because of its greater virulence, the likelihood of further PVS^II^ spread is a cause of concern for the biosecurity authorities of counties it has yet to reach. We find substantial genetic variation between the isolates within PVS^I^ and PVS^II^, but not within PVS^III^, and according to our neutrality tests, PVS^III^ isolates remain balanced whilst PVS^I^ and PVS^II^ exhibit expansion of their respective isolate populations. The high *F*_ST_ values assigned to PVS^I^, PVS^II^ and PVS^III^ comparisons is consistent with their separation into three major phylogroups. In addition, the ORF, RdRp, TGB and CP regions of PVS are under strong negative selection constraints. Eight of the major parents and six of the minor parents of the nine recombinant sequences belonged to PVS^I^, but one recombinant sequence had a recombinant major parent and three of them had minor parents that were either recombinants or belonged to PVS^II^. Moreover, our study expanded the geographical distribution of recombinant PVS sequences to include East Africa and East Asia in addition to Europe despite not finding any recombinant sequences amongst potato’s Andean domestication center where PVS originated. The twenty-two Turkish CP sequences belonged to five PVS^I^ subclades which is consistent with five separate incursions into the country, three of which likely represent recent arrivals. In our surveys of potato fields within Turkey, the incidence of PVS infection averaged 13% overall, and varied between incidences of 9% and 19% between individual provinces. This shows that PVS occurs commonly, so it is likely to be of economic importance for Turkish potato production.

### 4.2. Incursion History

When the origins of the 14 new CP sequences within PVS^I^ subclades 1–5 from our study, along with the 8 CP sequences of Engür and Topkaya [34], were mapped to different Turkish provinces, this information provided an insight into the likely history of the 5 PVS incursions into the country. Sequences from subclades 1 and 4 were each found in 3–4 of the 5 provinces, but subclades 2, 3 and 5 were only present in 1 province each (Figure 5). Therefore, the sequences within subclades 1 and 4 might represent 2 earlier PVS introductions that spread to other provinces subsequently, whereas, the sequences within subclades 2, 3 and 5 might represent more recent introductions yet to spread to other provinces. Moreover, since subclade 1, 2 and 4 isolate sequences grouped closely with Syrian and/or Iranian isolates, their introduction to Turkey might have occurred from these neighboring countries, rather than from more distant countries or from other continents. However, there was no evidence of this possible scenario with subclades 3 and 5, as their isolate sequences were closest to ones from countries much further away (Canada, China, and South Korea).

### 4.3. Recombination Findings

When Lin et al. 2014 [49] used the recombination program RDP4 to examine 44 nt sequences consisting of PVS’s CP and 11K genes (11K encodes an nt-binding protein), they reported detection of 19 potential recombination events by at least three of the seven recombination methods available. Santillan et al. [6] also used RDP4 when they tested 40 complete genomic PVS sequences. However, the level of stringency they applied was much higher as they ignored all anomalies not found by at least five of these same seven recombination methods. They reported presence of recombination events in five PVS genomic sequences, AJ863510, LN851189 and LN851192–4. Our study, which used the more sensitive RDP5 recombination program to examine many more (139) complete genomic PVS sequences, not only confirmed these five sequences as recombinants, but also found four others: one Kenyan (MN689463), one Japanese (LC375227) and two Chinese (KC430335, MK096268) (Table 2). The minor parent of European sequence AJ863510 was from the Bolivian Andes, but otherwise none of the Middle Eastern or Andean region sequences were recombinants themselves or parents of recombinants. Moreover, phylogenetic analysis of different coding regions of 130 recombinant-free PVS sequences generated trees with homogeneous topography (Figure 3), which substantiated their recombinant-free status.

### 4.4. Phylogenetics

When Santillan et al. [6] prepared phylogenetic trees of the non-recombinant PVS ORFs then available, phylogroup PVS^II^ was predominantly Andean, consisting of 10 new sequences they added from Andean region countries (Bolivia, Chile, Colombia, Ecuador, and Peru), but only 2 from elsewhere, 1 from Australasia (New Zealand) and 1 from East Asia (China). Similarly, after Duan et al. [5] added 10 new sequences from East Asia (China) to a phylogenetic tree of PVS CP sequences they published in the same year (2018), the only others their PVS^II^ contained were 3 Andean (Chile, Colombia) and 2 from non-Andean South America (Brazil). Over the 4–5 years period separating our PVS phylogenetic studies undertaken in 2022 from these two studies published in 2018 [5,6], our phylogenetic trees (ORF, RdRp, TGB, and CP; Figure 3 and Figure 4) revealed considerable differences in the geographical origins of the sequences within PVS^I^ and PVS^II^. PVS^I^ had become an increasingly ‘rest of world’ phylogroup. Although four Andean region sequences had been added to it by 2022, by that time many others from elsewhere were present from all continents except Antarctica (Africa, Australasia, Europe, North America, Southeast Asia, Middle East, and Central, South and East Asia). Notably, these included 45 sequences from just one African country (Kenya). Within PVS^II^, the biggest difference between 2018 and 2022 was the expansion of geographical origins of its non-South American sequences. This sequence expansion within PVS^II^ ranged from 11 sequences from East Asia (China) and 1 from New Zealand in 2018, to 12 from East Asia (China), 10 African (Kenya), 2 each from Australasia (New Zealand) and Europe (the Netherlands), and 1 from Central Asia (Kazakhstan) in 2022. Thus, more non-Andean sequences from different world regions were now present, especially from Africa and East Asia, making PVS^II^ global instead of being predominantly Andean. By contrast, PVS^III^ remained entirely Andean, despite a modest increase in its sequence numbers to six. Although the isolate RVC Andean (JX419379) sequence was first proposed as a distinct PVS lineage having a closer genetic relationship with PVS^I^ than PVS^II^ [21], our analysis of complete ORFs and three coding regions placed this sequence and five other isolate sequences from *S*. *phureja* within a separate phylogroup (PVS^III^). PVS^III^ had a closer genetic relationship with PVS^II^ than PVS^I^ and shared a basal node with PVS^II^ (Figure 3 and Figure 4).

As mentioned in the introduction, it makes no sense to retain the names PVS^O^ and PVS^A^ for groups of isolates defined both biologically and phylogenetically that do not coincide with their properties. Thus, there is no logic in using the name PVS^A^ (A = Andean) for PVS^II^ as many isolates outside the Andean region are now present within this phylogroup. Likewise, there is no logic in retaining the name PVS^RVC^ (this name is from isolate RVC) for PVS^II^, or the name PVS^O^ (O = ordinary) for PVS^I^ which now consists of isolates from all continents, except Antarctica. The likely reason for the expansion in both geographical distribution and numbers of PVS^II^ isolate sequences from outside the Andean region might be the importation of Andean potato germplasm inadvertently infected with PVS by potato breeding programs in other continents. Since the non-Andean countries of Kenya and China have the most new PVS^II^ sequences, this reflects the presence of highly active potato breeding programs in these two countries [50]. Such germplasm importation probably occurred mainly in the past when quarantine testing for viruses in potato germplasm dispatched as tubers were less stringent than present, and when virus sequencing technologies were less advanced and less virus sequencing was performed than present [51]. Although there are far fewer sequences within phylogroup PVS^III^ and all of these are currently Andean, more sequencing will likely reveal that it too has reached other continents. This might occur particularly via the distribution of germplasm consisting of *S. phureja* tubers, since currently, with one exception (MF4I8029 from *S. tuberosum* ssp. *andigena* referred above in the sections Introduction, and Materials and Methods) [6], this is the only potato species found so far infected with it (Figure 3 and Figure 4). Within PVS^I^, expansion of the international trade in seed potatoes between non-Andean countries seems the probable reason for the major diversification of global geographical isolate origins within this phylogroup. This is because this export trade consists mostly of seed tuber exports from Europe and North America to the rest of the world, with only a minor Andean region seed potato export contribution (from Chile which exports within South America) [52].

### 4.5. Population Genetics

When Lin et al. [49] undertook a population genetics study of 69 CP and 44 11K genes of PVS, they reported greater nt diversity (higher *π* value) with CP than 11K, suggesting greater variation in the CP gene. Additionally, they reported that PVS^I^ sequences had lower *π* values than PVS^II^ sequences, which was consistent with PVS^II^ having greater diversity. In addition, when they used three neutrality tests (Tajima’s *D*, Fu and Li’s *D**, and Fu and Li’s *F**), the values they obtained with both CP and 11K genes were positive for PVS^II^ but negative for PVS^I^. This suggested a balancing selection pressure on CP and 11K proteins for PVS^II^ but a negative selection pressure on them for PVS^I^. In our study, which was with many more sequences and which included complete ORF, RdRp and TGB regions in addition to CP sequences, the PVS^II^ population values for nt differences between sequences and nt diversity were consistently higher than those obtained with the PVS^I^ and PVS^III^ populations across all regions. In addition, there was considerable genetic variation amongst PVS^I^ and PVS^II^ isolate sequences, but low genetic variation within the PVS^III^ population. When the same three neutrality tests were used, PVS^I^ and PVS^II^ sequences always received negative values, but the PVS^III^ sequence values were always positive. This differs from the findings of Lin et al. [49] in which positive values were obtained with PVS^II^ sequences. This difference was presumably because they analyzed a much smaller number of PVS^II^ sequences (only 6) than we did (48), and a greater number of ours were from outside the Andean region. Our results indicate that both PVS^I^ and PVS^II^ are experiencing population expansion across all four genomic regions, but the smaller PVS^III^ population is undergoing balancing selection. Moreover, our *F*_ST_ values from comparisons between PVS^III^ versus either PVS^I^ or PVS^II^ at different genomic regions were always > 0.6 (Table 4), suggesting lack of gene flow, and expanding genetic separation between PVS^III^ and the other two phylogroups. This provides additional evidence that PVS^III^ is distinct. Furthermore, in our study, analysis of the sequences of 225 PVS isolates showed that the CP was the most conserved region, which was probably due to the strong purifying pressure on its short nt sequence. Thus, when sequences from other genomic regions are absent, the CP gene can still be used to represent PVS phylogeny. Other members of genus *Carlavirus* (potato virus M (PVM) [53] and garlic common latent virus (GCLV) [54]) also experience a strong negative evolutionary constraint on the CP. Thus, data obtained in this study could also contribute to other evolutionary studies of the genus *Carlavirus*.

### 4.6. Biosecurity Implications

Studies in Europe and North America reported that not only did more symptoms develop in *S. tuberosum* spp. *tubersosum* potato foliage with strain group PVS^A^, to which most phylogroup PVS^II^ isolates belong rather than to phylogroup PVS^I^, but also aphid transmission is more efficient with strain group PVS^A^ than with PVS^O^ [16,55,56,57]. Moreover, although symptomatology studies did not confirm this for phylogroup PVS^II^ when Andean *S. tuberosum* ssp. *andigena* or *S. tuberosum* ssp. *tuberosum* x *andigena* potato cultivars were used in South America, they did confirm it for PVS^II^ aphid transmission [6]. In addition, the South American studies [6] reported evidence that isolates within phylogroup PVS^II^ are not only more stable in infective sap than those in phylogroup PVS^I^, but also attain higher virion concentrations within infected plants, both properties likely to favor more efficient aphid and contact transmission in the field. Therefore, PVS^II^ is more likely to constitute a threat to seed and ware potato production for countries to which it is yet to spread, especially when co-infection then occurs with other viruses. Our study demonstrates that plant biosecurity and quarantine organizations in such countries should consider conducting surveillance programs to establish whether PVS^II^ has arrived, and taking additional precautions to help avoid its establishment. This is suggested as potato is the fourth most important staple food crop in the world and plays a major role in addressing food insecurity in developing countries [58,59].

## 5. Conclusions

We compared (i) all the non-recombinant sequences of complete ORF, RdRp and TGB available from GenBank, and (ii) the new CP’s we obtained from four Turkish provinces, and others obtained previously from a fifth province, with all other CP sequences in GenBank. Our comparisons involved recombination, phylogenetic and population genetic analyses. The more comprehensive sequence data set available now than in 2018, when the last genomic and CP analyses were conducted with this virus, enabled us to make important new deductions that helped build a better understanding of its evolution, the spread of its distinct phylogroups to other continents and the practical significance of this ongoing process. The suggested renaming of PVS’s three major phylogroups as PVS^I^, PVS^II^ and PVS^III^ was supported by recent spread of PVS^I^ and PVS^II^ around the world, especially the recent spread of PVS^II^ away from the Andean region of South America, where potato was first domesticated, to East Africa and East Asia. The genetic variation within phylogroups PVS^I^ and PVS^II^ was considerable, and both of their populations were expanding. By contrast, the PVS^III^ population had low variation and was undergoing balancing selection. The recombination involved found mostly parental sequences belonging to PVS^I^. As PVS^I^ infection was widespread in potato crops in Turkey’s potato producing regions, it is likely to have economic significance for the country, especially when it occurs in mixed infection with other viruses. The incursion history of the Turkish CP sequences within phylogroup PVS^I^ subclades 1–5 indicated presence of two earlier PVS introductions that spread to other provinces subsequently, and three more recent introductions yet to spread from the province in which they were found first. Further international spread of phylogroup PVS^II^ poses a biosecurity threat to world regions in which it is still absent. Such spread seems most likely to occur through distribution of germplasm for potato breeding purposes or further expansion of the international seed potato tuber trade.

## Figures and Tables

**Figure 1 viruses-15-01104-f001:**
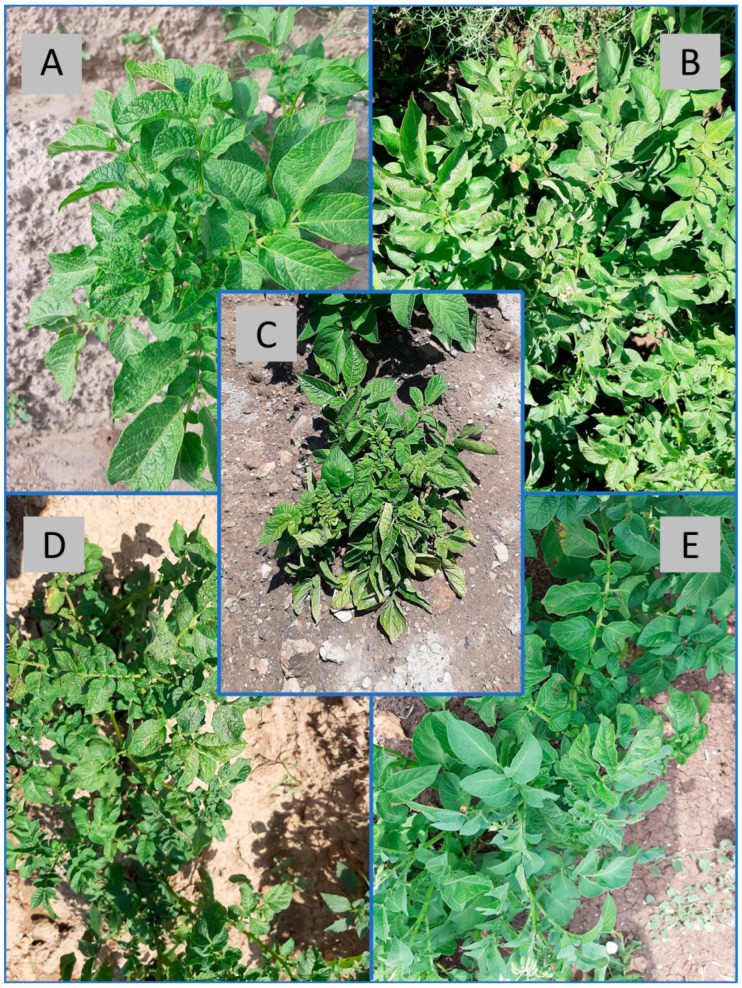
Sampled potato plants infected with potato virus S alone or in combination with other viruses. (**A**) Mild mosaic in young leaves of the infected plant from which isolate Ka1-7 was obtained; (**B**) Asymptomatic infection in the plant from which isolate SA12-5 was secured; (**C**) Leaf curling, mosaic, rugosity and chlorosis plus plant stunting in the plant from which isolate B9 was obtained; (**D**) Leaf curling, rugosity and mosaic in the plant from which isolate B14 was secured; (**E**) Leaf curling in young leaves of the infected the plant from which isolate Bo7 was obtained.

**Figure 2 viruses-15-01104-f002:**
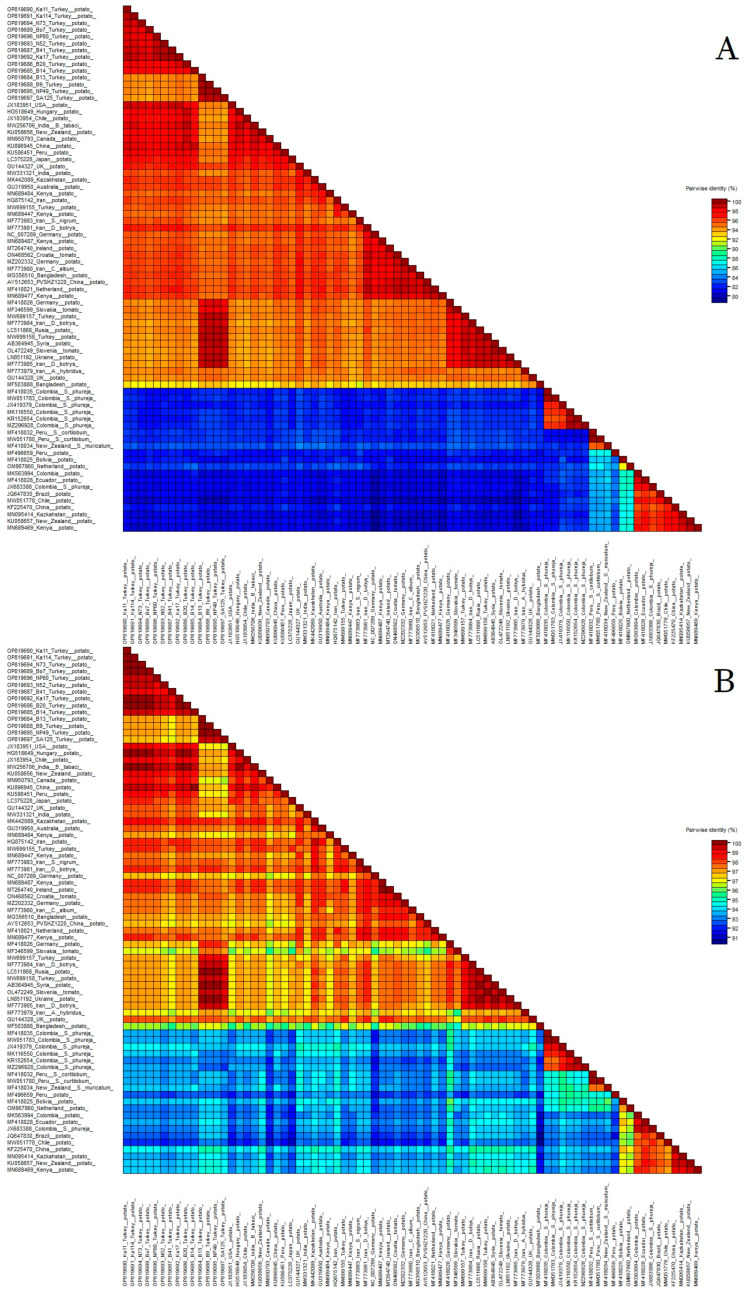
Percentage identities of the coat protein regions between 14 new Turkish Potato virus S isolates and 63 isolates from the NCBI GenBank. (**A**) nucleotide; (**B**) amino acid. Individual isolate names can be seen by increasing the magnification. Additionally, the phylogroup each isolate belonged to is shown in both Figure 3 and Appendix A.

**Figure 3 viruses-15-01104-f003:**
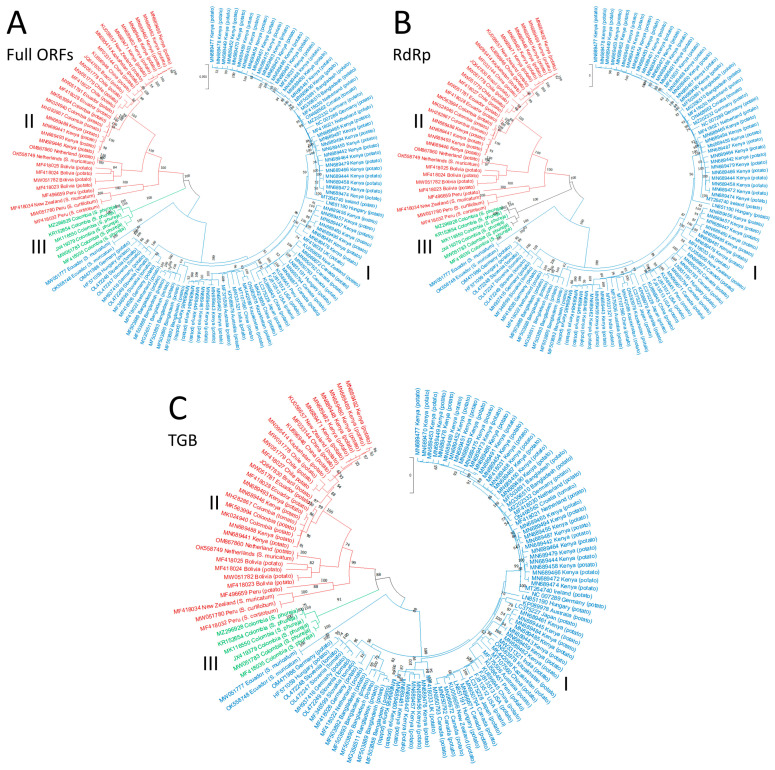
Phylogenetic trees based on the nucleotide sequences of three regions of potato virus S genome: (**A**) Complete ORFs; (**B**) RdRp; and (**C**) TGB. The trees were generated using the Tamura–Nei parameter model (TN93) in MEGA X software, with uniform rates among sites and 1000 bootstrap replicates (only values > 50% were shown). Different PVS phylogroups are highlighted with colors: PVS^I^ = blue, PVS^III^ = green, PVS^II^ = red.

**Figure 4 viruses-15-01104-f004:**
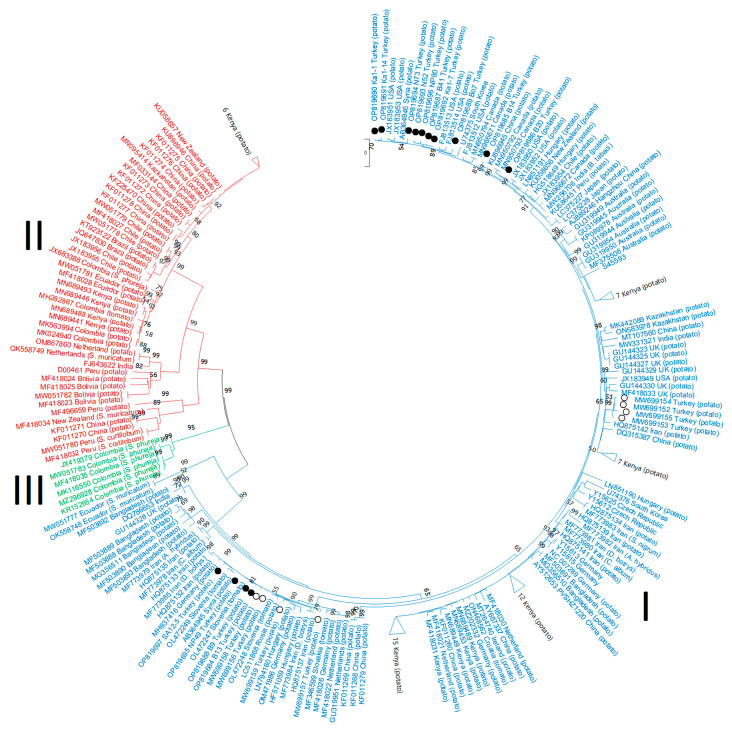
Phylogenetic trees based on the nucleotide sequences of the coat protein (CP) region of the potato virus S genome. The trees were generated using the Tamura–Nei parameter model (TN93) in MEGA X software, with uniform rates among sites and 1000 bootstrap replicates (only values > 50% were shown). Different PVS phylogroups are highlighted with colors: PVS^I^ = blue, PVS^III^ = green, PVS^II^ = red. The 14 new Turkish isolates are indicated with black circle symbols and the 8 Turkish isolates from Engür and Topkaya [34] with white circles.

**Figure 5 viruses-15-01104-f005:**
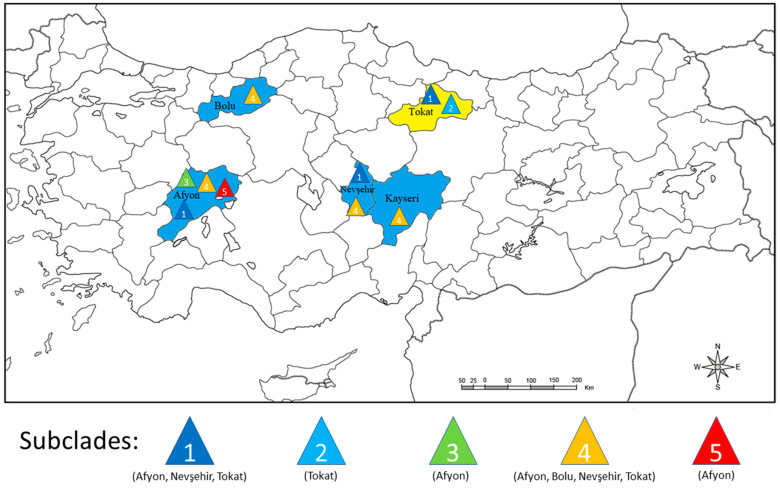
Distribution according to coat protein sequence of five different potato virus phylogroup PVS^I^ subclades within the five Turkish provinces sampled. Each colored triangle indicates a different PVS^I^ subclade. On the map, blue color identifies the four Turkish provinces where potato leaf samples were collected in this study, and yellow color the Turkish province sampled by Engür and Topkaya [34].

**Table 1 viruses-15-01104-t001:** Details of potato virus S (PVS) detections in potato samples collected from four Turkish provinces, isolates sequenced and accession numbers allocated.

Province	No. of Samples Collected	No. of Samples with PVS	No. of PVS Isolates Sequenced	Isolate Name (CP Accession no.) and (PVS^I^ Subclade)	Isolate Origins
Afyon	85	16	6	B13 (OP819684) (1) B9 (OP819688) (1) SA12-5 (OP819697) (1) B14 (OP819685) (3) B20 (OP819686) (5) B41 (OP819687) (4)	This study
Bolu	71	8	1	Bo7 (OP819689) (4)	This study
Kayseri	35	3	3	Ka1-1 (OP819690) (4) Ka1-7 (OP819692) (4) Ka1-14 (OP819691) (4)	This study
Nevşehir	73	8	4	N52 (OP819693) (4) N73 (OP819694) (4) NP80 (OP819696) (4) NP49 [OP819695] (1)	This study
Total	264	35	14	-	-
Tokat *	418	70	8	PA3-2 (MW699152) (2) PB5-4 (MW699153) (2) PB6-2 (MW699154) (2) PA7-1 (MW699155) (2) PN3-6 (MW699156) (1) PA3-3 (MW699157) (1) PN5-2 (MW699158) (1) PN14-3 (MW699159) (1)	[34]

* Eight PVS isolates from Tokat province sequenced by Engür and Topkaya [34] were also included in the molecular analysis.

**Table 2 viruses-15-01104-t002:** Putative recombination events detected in 9 out of 139 complete potato virus S genomes tested using RDP5 analysis.

No.	Recombinant	Parents: Major/Minor	Breakpoints ^1^ (Start/End)	RDP Implemented Method ^2^ (*p* Value)
1.	AJ863510 (Czech Republic)	MZ202332 (PVS^I^, Germany)/MW0517582 (PVS^II,^ Bolivia)	18/6055	R (3.287 × 10^−58^) G (1.037 × 10^−53^) B (1.387 × 10^−57^) M (8.144 × 10^−33^) C (1.961 × 10^−31^) S (9.952 × 10^−96^) 3S (7.898 × 10^−167^)
2.	KC430335 (China)	MN689457 (PVS^I^, Kenya)/FJ813513 (PVS^I^, USA)	4386/4814	R (5.887 × 10^−05^) G (8.432 × 10^−06^) B (1.295 × 10^−07^) M (1.488 × 10^−02^) S (4.501 × 10^−05^)
3.	LC375227 (Japan)	MK096268 (recombinant, China)/PVS^I^, LC375228 (Japan)	6857/7184	R (4.778 × 10^−16^) G (1.780 × 10^−15^) B (5.198 × 10^−19^) M (1.856 × 10^−08^) C (1.054 × 10^−06^) S (1.521 × 10^−14^) 3S (1.146 × 10^−07^)
4.	LN851189 (Ukraine)	MN689477 (PVS^I^, Kenya)/OL472247 (PVS^I^, Slovenia)	2626/8306	R (1.571 × 10^−24^) G (3.017 × 10^−19^) B (4.610 × 10^−25^) M (1.825 × 10^−21^) C (1.712 × 10^−02^) S (5.127 × 10^−37^) 3S (1.298 × 10^−25^)
5.	LN851192 (Ukraine)	OL472247 (PVS^I^, Slovenia)/MF418030 (PVS^I^, Netherlands)	64/2751	R (6.627 × 10^−32^) G (6.602 × 10^−24^) M (2.352 × 10^−25^) C (1.881 × 10^−26^) S (8.927 × 10^−25^) 3S (1.114 × 10^−72^)
6.	LN851193 (Ukraine)	OL472247 (PVS^I^, Slovenia)/MF418030 (PVS^I^, Netherlands)	64/2751	R (6.627 × 10^−32^) G (6.602 × 10^−24^) M (2.352 × 10^−25^) C (1.881 × 10^−26^) S (8.927 × 10^−25^) 3S (1.114 × 10^−72^)
7.	LN851194 (Poland)	MF418021 (PVS^I^, Netherlands)/LN851192 (recombinant, Ukraine)	5910/7132	R (6.083 × 10^−26^) G (1.320 × 10^−26^) B (6.687 × 10^−29^) M (1.101 × 10^−12^) C (2.457 × 10^−12^) S (4.668 × 10^−15^) 3S (8.473 × 10^−32^)
8.	MN689463 (Kenya)	MN689494 (PVS^I^, Kenya)/MN689448 (PVS^II^, Kenya)	21/311	G (5.316 × 10^−22^) B (1.761 × 10^−28^) M (2.354 × 10^−04^) C (2.273 × 10^−04^) S (3.638 × 10^−02^) 3S (4.644 × 10^−08^)
9.	MK096268 (China)	FJ813513 (PVS^I^, USA)/MK442089 (PVS^I^, Kazakhstan)	6241/6926	R (8.935 × 10^−06^) G (3.161 × 10^−07^) B (1.268 × 10^−08^) M (9.138 × 10^−10^) C (7.334 × 10^−04^) S (7.621 × 10^−23^) 3S (8.319 × 10^−04^)

^1^ Position in alignment; ^2^ R = RDP; G = GENECOV; B = BootScan; M = MaxChi; C = Chimaera; S = Siscan; 3 S = 3 Seq.

**Table 3 viruses-15-01104-t003:** Summary of genetic diversity and polymorphism analyses on four genomic regions from different potato virus S populations.

Phylogroups	*N*	*h*	*H_d_*	*S*	*η*	*k*	π	dS	dN	Ω
**Full ORFs**	130	128	0.999	4133	6308	1063.591	0.1278	0.3607	0.0593	0.1644
PVS^I^	91	89	1.000	3489	4439	474.948	0.0571	0.1668	0.0247	0.1481
PVS^III^	6	6	1.000	520	531	244.000	0.0293	0.0917	0.0109	0.1189
PVS^II^	33	33	1.000	2970	3724	784.453	0.0943	0.2775	0.0404	0.1456
**RdRP**	130	128	0.999	3045	4749	807.984	0.1363	0.4316	0.0502	0.1163
PVS^I^	91	89	1.000	2600	3334	359.116	0.0606	0.2028	0.0191	0.0942
PVS^III^	6	6	1.000	411	420	190.533	0.0321	0.1111	0.0091	0.0819
PVS^II^	33	33	1.000	2245	2840	599.011	0.1011	0.3363	0.0324	0.0963
**TGB**	130	123	0.999	541	750	118.844	0.0991	0.2647	0.0483	0.1825
PVS^I^	91	84	0.998	453	559	59.845	0.0499	0.1302	0.0252	0.1935
PVS^III^	6	6	1.000	45	48	23.133	0.0192	0.0513	0.0087	0.1696
PVS^II^	33	33	1.000	350	418	86.358	0.0719	0.1886	0.0364	0.1931
**CP**	224	210	0.999	435	687	93.011	0.1052	0.3759	0.0227	0.0603
PVS^I^	170	157	0.999	389	521	42.951	0.0486	0.1701	0.0115	0.0676
PVS^III^	6	6	1.000	53	53	25.533	0.0288	0.1081	0.0046	0.0426
PVS^II^	49	47	0.998	304	408	72.423	0.0818	0.2914	0.0179	0.0614

*N*—number of isolates, *h*—number of haplotypes, *Hd*—haplotype diversity, *S*—number of variable sites, *η*—total number of mutations, *k*—average number of nucleotide differences between sequences, π—nucleotide diversity (per site), dN—non-synonymous nucleotide diversity, dS—synonymous nucleotide diversity, Ω—dN/dS.

**Table 4 viruses-15-01104-t004:** Summary of demography test statistics for four genomic regions from different potato virus S populations.

Phylogroups	Fu and Li’s *D**	Fu and Li’s *F**	Tajima’s *D*
**Full ORFs**	0.58524 ns	0.21631 ns	−0.27604 ns
PVS^I^	−0.80344 ns	−1.36717 ns	−1.57175 ns
PVS^III^	0.22861 ns	0.27325 ns	0.32038 ns
PVS^II^	0.16750 ns	−0.10101 ns	−0.56069 ns
**RdRp**	0.70023 ns	0.30507 ns	−0.24821 ns
PVS^I^	−0.74118 ns	−1.31813 ns	−1.55871 ns
PVS^III^	0.14202 ns	0.17776 ns	0.23318 ns
PVS^II^	0.21777 ns	−0.06014 ns	−0.55630 ns
**TGB**	−0.00180 ns	−0.25618 ns	−0.45833 ns
PVS^I^	−1.00886 ns	−1.49813 ns	−1.56128 ns
PVS^III^	0.63604 ns	0.69883 ns	0.64308 ns
PVS^II^	−0.05321 ns	−0.29591 ns	−0.62058 ns
**CP**	−0.45046 ns	−0.62638 ns	−0.60418 ns
PVS^I^	−2.07196 ns	−2.26514 ns	−1.71837 ns
PVS^III^	0.55376 ns	0.62780 ns	0.64148 ns
PVS^II^	−0.18097 ns	−0.47960 ns	−0.76064 ns

ns = not significant.

**Table 5 viruses-15-01104-t005:** Genetic differentiation estimates for the major potato virus S phylogroups based on comparisons between sequences from four different genomic regions.

Comparison	^α^*K*_S_ *	^α^*K*_ST_ *	*p* Value	^α^*Z* *	*p* Value	*S* _nn_	*p* Value	^β^ *F* _ST_
**Complete ORFs**								
PVS^I^ (n = 91)/PVS^III^ (n = 6)	5.9673	0.0329	0.0000 ***	7.2663	0.0000 ***	1.0000	0.0000 ***	0.7975
PVS^I^ (n = 91)/PVS^II^ (n = 33)	6.0726	0.0794	0.0000 ***	7.4262	0.0000 ***	1.0000	0.0000 ***	0.6394
PVS^III^ (n = 6)/PVS^II^ (n = 33)	6.1672	0.0622	0.0000 ***	5.2196	0.0000 ***	1.0000	0.0000 ***	0.7019
**RdRp**								
PVS^I^ (n = 91)/PVS^III^ (n = 6)	5.6939	0.0342	0.0000 ***	7.2675	0.0000 ***	1.0000	0.0000 ***	0.7966
PVS^I^ (n = 91)/PVS^II^ (n = 33)	5.7986	0.0828	0.0000 ***	7.4271	0.0000 ***	1.0000	0.0000 ***	0.6392
PVS^III^ (n = 6)/PVS^II^ (n = 33)	5.9015	0.0641	0.0000 ***	5.2236	0.0000 ***	1.0000	0.0000 ***	0.6994
**TGB**								
PVS^I^ (n = 91)/PVS^III^ (n = 6)	3.8905	0.0498	0.0000 ***	7.2612	0.0000 ***	1.0000	0.0000 ***	0.7981
PVS^I^ (n = 91)/PVS^II^ (n = 33)	3.9838	0.1061	0.0000 ***	7.4409	0.0000 ***	1.0000	0.0000 ***	0.6054
PVS^III^ (n = 6)/PVS^II^ (n = 33)	3.9818	0.0957	0.0000 ***	5.1959	0.0000 ***	1.0000	0.0000 ***	0.7177
**CP**								
PVS^I^ (n = 170)/PVS^III^ (n = 6)	3.6536	0.0291	0.0000 ***	8.5398	0.0000 ***	1.0000	0.0000 ***	0.7892
PVS^I^ (n = 170)/PVS^II^ (n = 49)	3.7281	0.1127	0.0000 ***	8.6346	0.0000 ***	1.0000	0.0000 ***	0.6669
PVS^III^ (n = 6)/PVS^II^ (n = 49)	3.8668	0.0707	0.0000 ***	6.0309	0.0000 ***	1.0000	0.0000 ***	0.7005

*** *p* value < 0.001; ^α^*K*_S*_, *K*_ST*_, *Z** and *S*_nn_ are test statistics of genetic differentiation.; ^β^*F*_ST_, coefficient of gene differentiation, which measures inter-population diversity.

## Data Availability

Coat protein sequences of 14 novel Turkish PVS isolates have been made available in GenBank, reference numbers OP819684-OP819697.

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
