# Peer review of "Molecular Analysis of the Global Population of Potato Virus S Redefines Its Phylogeny, and Has Crop Biosecurity Implications"

_viruses, 2023, doi:10.3390/v15051104_

Round 1

Reviewer 1 Report

The study presents a well-written and informative molecular analysis of the Potato Virus S (PVS) global population. The materials and methods used in the study are appropriate and described in detail. The authors used high-throughput sequencing and phylogenetic analysis of the complete ORF, RdRp, and CP genes of different PVS isolates to characterize the genetic diversity and evolutionary history of PVS. The study is to provide a comprehensive overview of the molecular epidemiology of PVS, which is of great importance to the scientific community.

However, I have found several areas that could be improved.

1.        In the abstract line 26, the authors mentioned “PVSII spread constitutes a biosecurity threat for countries still free from it”. Need a bit of explanation of such a statement in the abstract. And in the last paragraph of discussion, same statement was made only by Santillan’ report, could the author provide any evidence from this study?

2.        The introduction section is too long and could be made more concise.

3.        The study's objectives are not clearly stated, making it difficult to understand the purpose of the research.

4.     How long were the samples stored in the 40C? It is not a safe temperature for samples using for RNA isolation. (See lines 136-137)

5.        In lines 147-149, the sentence should be clearer.

6.        How the authors had confirmed the mixed infections between PVS and other common potato viruses and what were the other viruses? Was there any test conducted, for example, ELISA for other virus detection? The addition of such information will help to understand more clearly and confirm the symptom was caused by PVS only or co-infection with other viruses (See lines 200-201 and Fig 1)

7.        It is understandable that the authors choose 14 PVS isolates for their CP region’s complete sequencing. However, was there any reason behind this selection? Table 1 it is mentioned that from Afyon’s 16 infected samples, 6 isolates were sequenced whereas, from Bolu’s 8 infected samples only 1 PVS Isolate was sequenced and from Kayseri’s, all 3 infected PVS isolates were sequenced. It does not look selecting randomly.

8.    The recombinants might be interesting for some readers, the authors should provide more information on this part, such as where the parents form, PVSI or PVSII, and any recombination hotspot among those ten recombinants, except for the geographical distribution (See lines 223-229 and Table2)

9.        Some discussion has been added in the result section, however, it can be included in the discussions section rather than the results section. (For example, see lines, 251-255 and 259-262)

10.    Lastly, there is no overall short summary of the findings of the study at the end of the discussion, which makes it difficult to understand the implications of the results.

11.    Line 69, CL should be CS; line 145, MgCl2 should be MgCl2; line 512, Cost protein should be Coat protein…

In conclusion, this study has the potential to contribute to our understanding of the molecular epidemiology of PVS, but some minor revisions are needed to improve the clarity and focus of the study.

Reviewer 2 Report

This is a nice piece of the work on the genetic diversity and evolution of potato virus S (PVS). To the best of my knowledge this is one of the first papers on this topic. The authors carried out bioinformatics analysis of all the Turkish PVS isolates based on the coat protein (CP) gene sequences and compared them with the PVS sequences available at NCBI. It would be more appropriate to analyse full length nucleotide sequences of Turkish PVS isolates rather than just the CP sequences. Moreover, it is not clear why the authors have decided to use the CP sequences, but not other virus genes for bioinformatics. This must be explained in the text. Due to these limitations description of Turkish isolates seems to be slightly marginal. However, in addition to this part of the work, the authors present more comprehensive analysis of global population of PVS using NCBI data, and thereby contribute further evidence justifying the new nomenclature of PVS strains consisting of the PVSI, PVSII and PVSIII groups. To summarize, I believe that the paper may be useful for many plant virologists and can be published in Viruses.

Reviewer 3 Report

The authors have performed the phylogenetic analysis to explore the PVS globally with some new data derived from several Turkish provinces. The authors revealed the PVS-II spread constitutes a biosecurity threat.

I believe that the authors have provided sufficient background, explained well the methodologies used, presented the data with appropriate tables and figures, and concluded appropriately based on available data. The language reads well, though there are occasionally minor grammatical and editorial errors. I have no major technical concerns but some minor recommendations listed below for authors to consider, if a revision is requested by the editor.

Line 33: is it necessary to include the latin name of potato? I believe that although there are numerous varieties of potato, only one species is recognized.

Line 115: the goals of this study should be explicitly described in this paragraph.

Figure 2: the labels are too small to be seen, is there any way to increase the resolution of the parts?

Either “MEGA-X” or “MEGA X” but be consistent.

Discussion:

I would highly recommend that the authors establish a few subsections, e.g., the different groupings of the viruses, to focus on the in-depth discussion of these clades.

Also, with the lengthy discussion, I would highly recommend that the authors establish a section of Conclusions to recap the major discoveries and their significance in this study.
